# Patch Gradient Descent: Training Neural Networks on Very Large Images

## Abstract

Traditional CNN models are trained and tested on relatively low resolution images (< 300 px), and cannot be directly used on large-scale images due to compute and memory constraints. We propose Patch Gradient Descent (PatchGD), an effective learning strategy that allows to train the existing CNN architectures on large-scale images in an end-to-end manner. PatchGD is based on the hypothesis that instead of performing gradient-based updates on an entire image at once, it should be possible to achieve a good solution by performing model updates on only small parts of the image at a time, ensuring that the majority of it is covered over the course of iterations. PatchGD thus extensively enjoys better memory and compute efficiency when training models on large scale images. PatchGD is thoroughly evaluated on two datasets - PANDA and UltraMNIST with ResNet50 and MobileNetV2 models under different memory constraints. Our initial evaluation reveals that PatchGD is much more stable and efficient than the standard gradient-descent method in handling large images, and especially when the compute memory is limited.

## 1   Introduction

In the realm of computer vision, Convolutional Neural Networks (CNNs) have established themselves as the cornerstone of advanced feature extraction, far surpassing traditional algorithms. Recent reviews by [1, 2, 3] encapsulate their evolution and dominance.

However, with the influx of high-dimensional data from sectors like microscopy [4, 5], medical imaging [6], and earth sciences [7, 8], the computational challenges for CNNs have surged. For example, high-content nanoscopy often necessitates the assimilation of multiscale data with information content relevant to the science present at scales ranging from a pixel to artifacts whose length-scales approach the image dimension − leading to issues in effective CNN application.

Most prevailing CNN models, fine-tuned on datasets such as ILSVRC and PASCAL VOC, which mainly comprise of low-resolution (< 300 pixels) images, encounter difficulties when extended to high-resolution images due to dramatic increase in intermediate activations. Common mitigative strategies—like downsampling or tiling—either compromise the feature fidelity or disrupt contextual continuity. Attention mechanisms, while providing semantic continuity, are often computationally prohibitive for high-res data due to their quadratic dependence on input token lengths.

Addressing this, we propose a robust CNN training paradigm tailored for high-dimensional data. The term "large" in our context is fluid, contingent on the computational memory overhead. For illustration, a $10,000 \times 10,000$ image might overextend a 48 GB GPU, but a $512 \times 512$ one is manageable on 12 GB—though the latter becomes challenging at a leaner 4 GB constraint. An example experimental demonstration on UltraMNIST digits [9] is presented in Figure 1. Herein

Submitted to the Workshop on Advancing Neural Network Training at 37th Conference on Neural Information Processing Systems (WANT@NeurIPS 2023). Do not distribute.

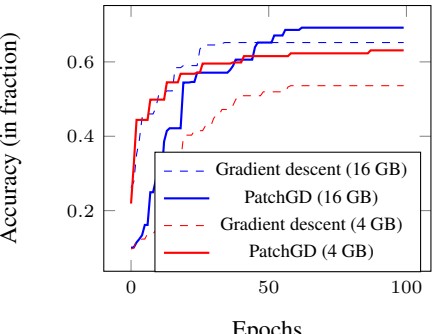

Figure 1: Performance comparison of standard CNN and PatchGD (ours) for the task of classification of UltraMNIST digits of size $512 \times 512$ pixels using ResNet50 model. Two different computational memory budgets of 16 GB and 4GB are used, and it is demonstrated that PatchGD is relatively stable for the chosen image size, even for very low memory compute.

lies the significance of our Patch Gradient Descent (PatchGD), demonstrating resilience across two different budget constraints.

**Contributions.** To summarize, the contributions of this paper can be listed as follows.

- We present *Patch Gradient Descent (PatchGD)*, a novel strategy to train neural networks on very large images in an end-to-end manner. PatchGD is an adaptation of the conventional feedforward-backpropagation optimization framework.

- Due to its inherent ability to work with small fractions of a given image, PatchGD is scalable on small GPUs, where training the original full-scale images may not even be possible.

- PatchGD reinvents the existing CNN training pipeline in a very simplified manner and this makes it compatible with any existing CNN architecture or any conventional gradient-based optimization method used in deep learning. Moreover, its simple design allows it to benefit from the pre-training of the standard CNNs on low-resolution data.

## 2 Approach

### 2.1 General description

*Patch Gradient Descent (PatchGD)* is a novel CNN training strategy that can train networks with high-resolution images. An adaptation of the standard feedforward-backpropagation method, it is based on the hypothesis that, rather than performing gradient-based updates on an entire image at once, it is possible to achieve a good solution by performing model updates on only small parts of the image at a time, ensuring that the majority of it is covered over the course of iterations. However, even if only a portion of the image is used, the model is still trainable end-to-end with PatchGD.

In Figure 2, the PatchGD approach is presented schematically. The central idea behind PatchGD is to construct the $\mathbf{Z}$ block, which is a deep latent representation of the entire input image. Although only a subset of the input is used to perform model updates, $\mathbf{Z}$ captures information about the entire image by combining information from different parts of the image acquired from the previous update steps. Figure 2a illustrates the use of the $\mathbf{Z}$ block, which is an encoding of an input image $\mathbf{X}$ using a model parameterized by weights $\boldsymbol{\theta}_1$. The input image is divided into patches of size $m \times n$, and each patch is processed independently using $\boldsymbol{\theta}_1$. The size of $\mathbf{Z}$ is always enforced to be $m \times n \times s$, such that each patch in the input space corresponds to the respective $1 \times 1 \times s$ segment in the $\mathbf{Z}$ block.

The filling of $\mathbf{Z}$ is carried out in multiple steps, with each step involving the sampling of $k$ patches along with their positions from $\mathbf{X}$ and feeding them to the model as a batch for processing. The output from the model along with the corresponding positions are then used to fill the respective parts of $\mathbf{Z}$. After sampling all $m \times n$ patches of $\mathbf{X}$, the completely filled $\mathbf{Z}$ is obtained. This concept of $\mathbf{Z}$-filling is utilized by PatchGD during both training and inference stages. To create an end-to-end CNN model, we incorporate a small subnetwork that consists of convolutional and fully-connected layers. This subnetwork processes the information contained in $\mathbf{Z}$ and converts it into a $c$-dimensional probability vector, which is essential for the classification task. It is worth noting that the computational cost of adding this small subnetwork is minimal. The Figure 2b illustrates the pipelines for both model training and inference stages. During training, the model components $\boldsymbol{\theta}_1$ and $\boldsymbol{\theta}_2$ are updated. We compute the respective encodings based on a fraction of the patches sampled from the input image, using the latest state of $\boldsymbol{\theta}_1$, and update the corresponding entries in the already filled $\mathbf{Z}$ using the

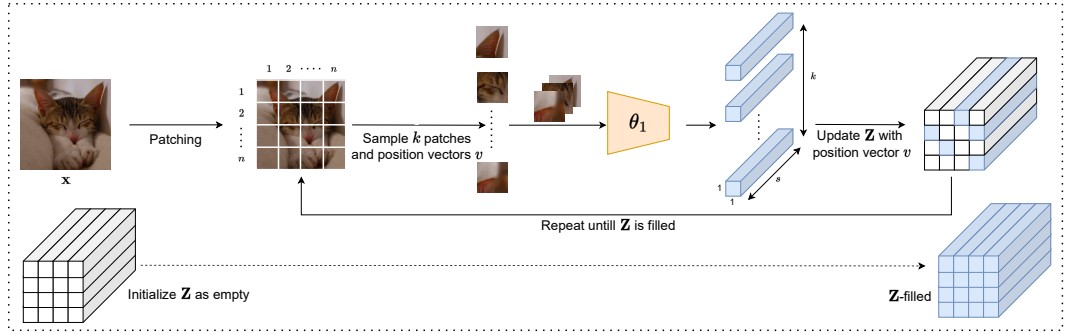

(a) Pipeline for the filling of **Z** block, also referred as **Z**-filling.

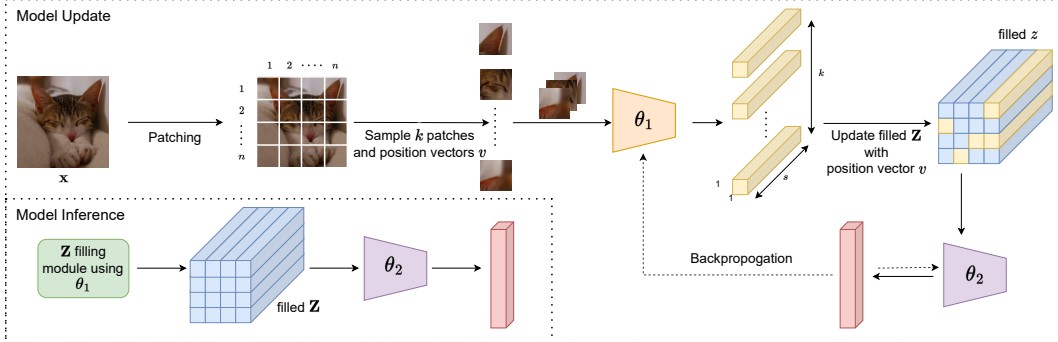

(b) Model update and model inference.

Figure 2: Schematic representations of the pipelines demonstrating working of different components of the PatchGD process.

model output. Subsequently, we use the partially updated **Z** to calculate the loss function value and update the model parameters using backpropagation. For more details, see the mathematical formulation presented in Appendix B.

# 3 Experiments

We showcase the effectiveness of PatchGD through numerical experiments on two benchmark datasets with large images and multiple scales, and additional experiments on generative modelling.

## 3.1 Results

Table 1: Performance scores obtained using Resnet50 on PANDA dataset for Gradient Descent (GD) and Patch Gradient Descent (PatchGD).

| Method | Resolution | Patch Size | Batch Size | Mem. (GB) | Throughput (imgs/sec) | Accuracy % | QWK |
|--------|-----------|-----------|-----------|-----------|----------------------|-----------|-----|
| Baseline | 512 | - | 27 | 16 | 618.05 | 44.4 | 0.558 |
| PatchGD | 512 | 128 | 86 | 16 | 521.42 | 44.9 | 0.576 |
| PatchGD | 512 | 64 | 200 | 16 | 341.87 | 52.1 | 0.616 |
| Baseline | 2048 | - | 1 | 16 | 39.04 | 34.8 | 0.452 |
| PatchGD | 2048 | 128 | 14 | 16 | 32.52 | 53.9 | 0.627 |
| Baseline | 2048 | - | 6 | 48 | 39.04 | 49.4 | 0.625 |
| PatchGD | 2048 | 128 | 56 | 48 | 32.52 | 56.2 | 0.667 |
| Baseline | 4096 | - | 1 | 48 | 9.23 | 50.0 | 0.611 |
| PatchGD | 4096 | 256 | 26 | 48 | 9.62 | 59.7 | 0.730 |

**UltraMNIST classification.** The performance of PatchGD for UltraMNIST has already been shown in Figure 1. PatchGD improves over the standard gradient descent method (abbreviated as GD) by large margins. The performance difference is even higher when we have a low memory constraint.

At 4 GB, while GD seems unstable with a performance dip of more than 11% compared to the 16 GB case, our PatchGD approach seems to be significantly more stable. The underlying reason for this gain can partly be attributed to the fact that since PatchGD facilitates operating with partial images, the activations are small and more images per batch are permitted.

**Prostate Cancer Classification (PANDA).** Table 1 presents the results obtained on PANDA dataset for three different image resolutions. For all experiments, we maximize the number of images used per batch while also ensuring that the memory constraint is not violated. For images of $512 \times 512$, we see that PatchGD, with patches of size $128 \times 128$, delivers approximately the same performance score as GD (for both accuracy as well as QWK) at 16 GB memory limit. However reducing the patch size and thus increasing the batch size, we observe a very sharp gain in the scores of PatchGD. For a similar memory constraint, when images of size $2048 \times 2048$ pixels are used, the performance of GD drops by approximately 10% while our PatchGD shows a boost of 9% in accuracy.

Two factors contribute to the performance gap between GD and PatchGD. Firstly, GD faces a bottleneck with batch size due to increased activation size in higher-resolution images, allowing only 1 image per batch. Gradient accumulation across batches and hierarchical training were explored but did not improve performance significantly. Increasing the memory limit helped mitigate the issue of using only 1 image per batch. Secondly, the optimized receptive field of ResNet50 is not well-suited for higher-resolution images, resulting in suboptimal performance. PatchGD demonstrates superior accuracy and QWK compared to GD on the PANDA dataset when handling large images end-to-end. In terms of inference latency, PatchGD performs comparably to GD. The smaller activations in PatchGD offset the slowness caused by patchwise image processing. PatchGD shows potential for real-time inference in applications requiring large image handling.

**Comparison with existing methods.** We further present a comparison of PatchGD with the existing methods designed for handling large images, and the results are presented in Table 2 of the appendices. Note that almost all works that exist on handling large images are not designed to work with memory constraints, and if put in such applications, these lead to unstable performance scores. For example, although the vision transformer backbones of HIPT are pretrained on large medical datasets, the performance of the model in the memory-constrained setting is lowest among the 4 methods presented in the table. For HIPT, all the layers of the vision transformer backbones are trainable and a batch size of only 5 fits in the memory. The original HIPT model is trained with large batch sizes over a set of GPUs, however, in our memory-constrained set up, it is not possible. The performance of ABNN and C2C is relatively better, however, they are still significantly lower than the PatchGD training of a simple architecture. C2C employs attention modules in the head of the network, and we believe with such additions, the performance of PatchGD could be boosted even further. Nevertheless, we see from the presented results that for memory-constrained settings, PatchGD performs significantly better than any other existing method when it comes to handling large images.

For HIPT, We conducted an additional experiment with gradient accumulation over 12 steps, referred as HIPT-L in Table 2. This led to an equivalent batch size of 60. Although the convergence was slow, the performance of the model boosted from 34.8 to 49.3. This clearly demonstrates that transformers with gradient accumulation could work well even at low batch sizes. Nevertheless, we still see a significant performance gap of more than 10% between HIPT and our approach. Moreover, transformers are known to be data hungry and one important thing to note here is that the pre-trained HIPT model we are using in this paper is already heavily trained on a very large medical dataset comprising training images from a variety of medical datasets. On the contrary, our model is only pre-trained on standard ImageNet and no additional pre-training is done. This clearly makes our approach stand out when compared to HIPT in the sense that it is applicable for low memory as well as relatively low training data regimes as well.

# 4 Conclusions

In this paper, we introduced Patch Gradient Descent (PatchGD), a novel CNN training strategy that effectively handles large images even with limited GPU memory. PatchGD updates the model using partial image fractions, ensuring comprehensive context coverage over multiple steps. Through various experiments, we demonstrated the superior performance of PatchGD compared to standard gradient descent, both in handling large images and operating under low memory conditions. The presented method and experimental evidence highlight the significance of PatchGD in enabling existing CNN models to effectively process large images without compute memory limitations.

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

# A  Additional results

We presented in Table 2 a comparison with existing works.

**Generative modelling and other tasks.** PatchGD can be used for generating large-scale images with a broad semantic context, which can be beneficial for data augmentation in fields such as deep learning for medical imaging. Early results using StyleGAN-2 on the CIFAR-10 dataset showed that our method generated patches of $16 \times 16$ which were stitched together and analyzed by the discriminator, leading to a comparable FID score of 6.3 to the standard GD's FID score of 6.1. We

Table 2: Comparison with existing methods at 4096 image size and 48GB memory constraint.

| Method | Accuracy % | QWK |
|---|---|---|
| HIPT [10] | 34.8 | 0.388 |
| HIPT-L | 49.3 | 0.531 |
| ABNN [11] | 48.2 | 0.593 |
| C2C [12] | 50.9 | 0.668 |
| PatchGD | **59.7** | **0.730** |

believe this small performance gap can be eliminated with hyperparameter optimization. We consider that the potential of PatchGD in generative modeling can be maximized by generating large images with various semantic contexts, although this needs to be explored further.

**On using PatchGD with transformers.** The current implementation of PatchGD has a limitation when it comes to using transformer architectures. From our investigation, we have observed that when using a DeiT backbone with PatchGD, the model performance is significantly inferior. This reveals that CNNs are a better choice of current PatchGD implementations. Our intuition is that the classification token in transformers strongly relies on seeing the full image from the start (through connection between patches), however, since this is not true when coupled with PatchGD, the performance of PatchGD with DeiT deteriorates. The current implementation of PatchGD assumes that the distant patches in the image are completely independent from each other, and the first connection of the patches happens at the L1-block; before that each patch is treated as an independent image. While it works for CNNs, this assumption does not hold for transformers. For transformers, information flow between patches happens right from the beginning, and mixing happens at every block. Clearly, the best way to build a PatchGD pipeline for transformers is to have something similar to L1 construction after every block. However, with such an approach, one needs to calculate gradients repeatedly for every block for only a fraction of the patches and approximate the rest of the history. We have considered this as part of the ongoing extension of this work so that PatchGD can be efficiently coupled with transformers.

# B    Mathematical formulation

In this section, we present a detailed mathematical formulation of the proposed PatchGD approach and describe its implementation for the model training and inference steps. For the sake of simplicity, we tailor the discussion towards the training of a CNN model for the task of classification.

Let $f_{\boldsymbol{\theta}} : \mathbb{R}^{M \times N \times C} \to \mathbb{R}^c$ denote a CNN-based model parameterized by $\boldsymbol{\theta}$ that takes an input image $\mathbf{X}$ of spatial size $M \times N$ and $C$ channels and computes the probability of it to belong to each of the $c$ pre-defined classes. To train this model, the following optimization problem is solved.

$$\min_{\boldsymbol{\theta}} \ \mathcal{L}(f(\boldsymbol{\theta}; \mathbf{X}), \mathbf{y}), \tag{1}$$

where $\mathbf{X}, \mathbf{y} \in \mathcal{D}$ represents the data samples used, and $\mathcal{L}(\cdot)$ represents the loss function. The conventional approach in deep learning is to solve this problem using mini-batch gradient descent, where updates are made using a subset of the data samples at each step. Below, we provide the formulations for standard gradient descent and our PatchGD method.

**Gradient Descent (GD).** Gradient descent in deep learning involves performing model updates using the gradients computed for the loss function over one or more image samples. With updates performed over one sample at a time, referred to as the stochastic gradient descent method, the model update at the $i^{\text{th}}$ step can be mathematically stated as

$$\boldsymbol{\theta}^{(i)} = \boldsymbol{\theta}^{(i-1)} - \alpha \frac{\mathrm{d}\mathcal{L}}{\mathrm{d}\boldsymbol{\theta}^{(i-1)}}, \tag{2}$$

where $\alpha$ denotes the learning rate. However, performing model updates over one sample at a time leads to very slow convergence, especially because of the noise induced by the continuously changing descent direction. This issue is alleviated in the mini-batch gradient descent method where at every step, the model weights are updated using the average of gradients computed over a batch of samples,

**Algorithm 1** Model Training for 1 iteration
___
1: **Input:** Batch of input images $\mathcal{X} \in \mathbb{R}^{B \times M \times N \times C}$, Pre-trained feature extractor $f_{\boldsymbol{\theta}_1}$, Classifier head $g_{\boldsymbol{\theta}_2}$, Patch size $p$, Inner iterations $\zeta$, Patches per inner iteration $k$, Batch size $B$, Learning rate $\alpha$, Grad. Acc. steps $\epsilon$
2: **Initialize:** $\mathbf{Z} \leftarrow \mathbf{0}^{B \times m \times n \times c}, \mathbf{U}_1 \leftarrow \mathbf{0}, \mathbf{U}_2 \leftarrow \mathbf{0}$
3: $\mathbf{Z} \leftarrow \mathbf{Z}\text{-filling}(\mathbf{X}, f_{\boldsymbol{\theta}_1}, p)$ forall $\mathbf{X} \in \mathcal{X}$
4: $f_{\boldsymbol{\theta}_1} \leftarrow \texttt{start\_gradient}(f_{\boldsymbol{\theta}_1})$
5: **for** $j : 1$ to $\zeta$ **do**
6:     **for** $\mathbf{X}$ in $\mathcal{X}$ **do**
7:        $\{\mathcal{P}_{\mathbf{X},j}, v\} \leftarrow \texttt{patch\_sampler}(\mathbf{X}, k)$,
8:        $\mathcal{P}_{\mathbf{X},j} \in \mathbb{R}^{p \times p \times C \times k}$
9:        $\mathbf{z} \leftarrow f_{\boldsymbol{\theta}_1}(\mathcal{P}_{\mathbf{X},j})$
10:       $\mathbf{Z}[v] \leftarrow \mathbf{z}$ // Update the positional embeddings
11:       $\mathbf{y}_{\text{pred}} \leftarrow g_{\boldsymbol{\theta}_2}(\mathbf{Z})$
12:       $\mathcal{L} \leftarrow \texttt{calculate\_loss}(\mathbf{y}, \mathbf{y}_{\text{pred}})$
13:       $\mathbf{U}_1 \leftarrow \mathbf{U}_1 + \mathrm{d}\mathcal{L}/\mathrm{d}\boldsymbol{\theta}_1, \mathbf{U}_2 \leftarrow \mathbf{U}_2 + \mathrm{d}\mathcal{L}/\mathrm{d}\boldsymbol{\theta}_2$
14:     **end for**
15:     **if** $j\%\epsilon = 0$ **then**
16:       $\mathbf{U}_1 \leftarrow \mathbf{U}_1/\epsilon, \mathbf{U}_2 \leftarrow \mathbf{U}_2/\epsilon$
17:       $\boldsymbol{\theta}_1 \leftarrow \boldsymbol{\theta}_1 - \alpha \mathbf{U}_1$
18:       $\boldsymbol{\theta}_2 \leftarrow \boldsymbol{\theta}_2 - \alpha \mathbf{U}_2$
19:       $\mathbf{U}_1 \leftarrow \mathbf{0}, \mathbf{U}_2 \leftarrow \mathbf{0}$
20:     **end if**
21: **end for**=0
___

**Algorithm 2** Filling of the $\mathbf{Z}$ block (referred as $\mathbf{Z}$-filling)
___
**Input:** Input image $\mathbf{X} \in \mathbb{R}^{M \times N \times C}$, Pre-trained feature extractor $f_{\boldsymbol{\theta}_1}$, Patch size $p$, $n \leftarrow (N/p), m \leftarrow (M/p)$
**Initialize:** $\mathbf{Z} \in \mathbb{R}^{m \times n \times s}, \boldsymbol{\theta}_1 \leftarrow \texttt{stop\_graph}(\boldsymbol{\theta}_1)$
**repeat**
    $\mathbf{x}_{a,b} \leftarrow \texttt{patch\_extractor}(\mathbf{X}, a, b)$
    $\mathbf{x}_{a,b} \in \mathbb{R}^{p \times p \times C}$
    $\mathbf{z}_{a,b} \leftarrow f_{\boldsymbol{\theta}_1}(\mathbf{x}_{a,b}), \mathbf{z}_i \in \mathbb{R}^{1 \times 1 \times s}$
    $\mathbf{Z}[a, b] \leftarrow \mathbf{z}_{a,b}$
**until** all patches sampled
**Return** $\mathbf{Z}$ =0
___

denoted here as $\mathcal{S}$. Based on this, the update can be expressed as

$$\boldsymbol{\theta}^{(i)} = \boldsymbol{\theta}^{(i-1)} - \frac{\alpha}{N(\mathcal{S})} \sum_{\mathbf{X} \in \mathcal{S}} \frac{\mathrm{d}\mathcal{L}^{(\mathbf{X})}}{\mathrm{d}\boldsymbol{\theta}^{(i-1)}} \tag{3}$$

and $N(S)$ here denotes the size of the batch used. As can be seen in Eq. 3, if the size of image samples $s \in \mathcal{S}$ is very large, it will lead to large memory requirements for the respective activations, and under limited compute availability, only small values of $N(\mathcal{S})$, sometimes even just 1 fits into the GPU memory. This should clearly demonstrate the limitation of the gradient descent method when handling large images. This issue is alleviated by our PatchGD approach and we describe it next.

**PatchGD.** As described in Section 2.1, PatchGD avoids model updates on an entire image sample in one go, rather it computes gradients using only part of the image and updates the model parameters. In this regard, the model update step of PatchGD can be stated as

$$\boldsymbol{\theta}^{(i,j)} = \boldsymbol{\theta}^{(i,j-1)} - \frac{\alpha}{k \cdot N(\mathcal{S}_i)} \sum_{\mathbf{X} \in \mathcal{S}_i} \sum_{p \in \mathcal{P}_{\mathbf{X},j}} \frac{\mathrm{d}\mathcal{L}^{(\mathbf{X},p)}}{\mathrm{d}\boldsymbol{\theta}^{(i,j-1)}}. \tag{4}$$

Here, $i$ here refers to a mini-batch iteration within a certain epoch. Further, $j$ denotes the inner iterations, where at every inner iteration, $k$ patches are sampled from the input image $\mathbf{X}$ (denoted as $\mathcal{P}_{\mathbf{X},j}$) and the gradient-based updates are performed as stated in Eq. 4. Note that for any iteration $i$, multiple inner iterations are run ensuring that the majority of samples from the full set of patches that are obtained from the tiling of $\mathbf{X}$ are explored.

In Eq. 4, $\boldsymbol{\theta}^{(i,0)}$ denotes the initial model for the inner iterations on $\mathcal{S}_i$ and is equal to $\boldsymbol{\theta}^{(i-1,\zeta)}$, the final model state after $\zeta$ inner iterations of patch-level updates using $\mathcal{S}_{i-1}$. For a more detailed understanding of the step-by-step model update process, please see Algorithm 1. As described earlier, PatchGD uses an additional sub-network that looks at the full latent encoding $\mathbf{Z}$ for any input image $\mathbf{X}$. Thus the parameter set $\boldsymbol{\theta}$ is extended as $\boldsymbol{\theta} = [\boldsymbol{\theta}_1, \boldsymbol{\theta}_2]^\intercal$, where the base CNN model and the additional sub-network are $f_{\boldsymbol{\theta}_1}$ and $g_{\boldsymbol{\theta}_2}$, respectively.

Algorithm 1 describes model training over one batch of $B$ images, denoted as $\mathcal{X} \in \mathbb{R}^{B \times M \times N \times C}$. As the first step of the model training process, $\mathbf{Z}$ corresponding to each $\mathbf{X} \in \mathcal{X}$ is initialized. The process of filling of $\mathbf{Z}$ is described in Algorithm 2. For patch $\mathbf{x}_{ab}$, the respective $\mathbf{Z}[a, b, :]$ is updated using the output obtained from $f_{\boldsymbol{\theta}_1}$. Note here that $\boldsymbol{\theta}_1$ is loaded from the last state obtained during the model update on the previous batch of images. During the filling of $\mathbf{Z}$, no gradients are stored for backpropagation.

Next, the model update process is performed over a series of $\zeta$ inner-iterations, where at every step $j \in \{1, 2, \ldots, \zeta\}$, $k$ patches are sampled per image $\mathbf{X} \in \mathcal{X}$ and the respective parts of $\mathbf{Z}$ are updated. Next, the partly updated $\mathbf{Z}$ is processed with the additional sub-network $\boldsymbol{\theta}_2$ to compute the class probabilities and the corresponding loss value. Based on the computed loss, gradients are backpropagated to perform updates of $\boldsymbol{\theta}_1$ and $\boldsymbol{\theta}_2$. Note that we control here the frequency of model updates in the inner iterations through an additional term $\epsilon$. Similar to how a batch size of 1 in mini-batch gradient descent introduces noise and adversely affects the convergence process, we have observed that gradient update per inner iteration leads to sometimes poor convergence. Thus, we introduce gradient accumulation over $\epsilon$ steps and update the model accordingly. Note that gradients are allowed to backpropagate only through those parts of $\mathbf{Z}$ that are active at the $j^{\text{th}}$ inner-iteration. During inference phase, $\mathbf{Z}$ is filled using the optimized $f_{\boldsymbol{\theta}_1^*}$ as described in Algorithm 2 in supplementary material, and then the filled version of $\mathbf{Z}$ is used to compute the class probabilities for input $\mathbf{X}$ using $g_{\boldsymbol{\theta}_2^*}$.

