# OpenReview forum: "Patch Gradient Descent: Training Neural Networks on Very Large Images"
_NeurIPS.cc/2023/Workshop/WANT — WANT@NeurIPS 2023 Poster_

### Official Review · Reviewer_Bi9K · 2023-10-23
**Good idea though the conduct of research is poor**

**Confidence:** 5

**Review:**

The patch-gradient decent procedure introduced by authors and considered in applications to image classification tasks assumes that a CNN network is designed in a way that, both during the training and inference, it sees only a random sub-set of non-overlapping patches of each image in a dataset. This, as the authors claim, is sufficient to achieve results compatible to networks of similar architecture trained with conventional GD.

Processing images by patches is not a novel idea: like in DeiT or ViT (mentioned in the paper), PatchGD splits an image into non-overlapping patches; a random subset of patches or all of them are further processed with shared weights. This is similar to how PatchGAN works (though patches there may overlap) ­— **not mentioned** in the paper under review. In pix2pix paper, Isola et al. investigated similar issue: how the patch size used in a discriminator for capturing local style statistics affects the quality of a generator.

In my opinion, the concept of the CNN presented in PatchGD is a mix of heuristics used in DeiT (non-overlapping patches, Z block is a more precise version of positional encodings) and PatchGAN (shared weights). Nevertheless, the idea of training a ANN on the partial content of images is new. I was surprised to find higher results in the experiments conducted by the authors. Since I expected that extracting a random set of patches would result in mislabeling data, thus, deteriorating the performance.

However, there are some concerns about the obtained results. It is not explicitly stated in the paper but from the results it can be deduced that ResNet50 was pre-trained in the memory-limited constraints on a dataset of high resolution images. In the light of that, it would have been reasonable to substitute BatchNorm layers with instance normalization, since the former performs poorly on small batches, especially when it is just one image. Not taking this into account introduces a bias when checking the main hypothesis: whether CNNs can be trained on image partial content

Summary of the review:
- Unreliable results in my opinion
- Vaguely formulated contributions (second and third points in the contributions are corollaries of the fist one)
- Poor overview of related works

---

### Official Review · Reviewer_CgKc · 2023-10-24
**The authors propose a pipeline able to do gradient descent on parts of the image. This allows to train using fewer memory. Their preliminary experiments demonstrate promising results: they outperform the SOTA in both performance and efficiency. Despite many unexplained details, the proposed method opens a door for future research.**

**Confidence:** 3

**Review:**

# Summary of the paper

The authors propose a patching method to train neural networks. At each iteration, the model only sees randomly selected parts of the image. They do so by training a latent representation of the image, in which each patch of the image is encoded into a latent vector. The latent representation is partially updated at each iteration. They add another convolutional network that takes this representation as input, and outputs a vector of probability.

The method allows to train with fewer ressources as the training only needs to see parts of the image. However, the training may take longer to converge. The authors demonstrate its efficiency on preliminary evaluations where it outperforms SOTA methods, with more efficient resources.

# Strengths

- The paper is well written and easy to follow.
- The proposed method is simple and opens the door for future research.
- The results on the preliminary evaluations are promising. They both outperform the SOTA and their baselines by a large margin, and do so with fewer resources needed.

# Weaknesses

- Many details are left unexplained (see Questions section). This is mainly due to the choice of having a 4-page paper. Why not proposing a full paper?
- The evaluation is a bit shallow. Again, this is due to having a 4-page paper.

# Remarks

- The authors say that the usual networks are trained on <300px. However, even MNIST images have 28x28 = 784 pixels. I am sure the authors meant 300x300 px: I would propose to change to <300x300 px for clarity.
- Lines 26-28, on common mitigative strategies, would deserve a reference for more information.
- Please add a small explanation of QWK as this metric is not often used

# Questions

- The authors say in the abstract that they ensure the majority of the image is seen throughout the training. How is this ensured? How are the patches sampled ?
- The authors say that they use a ResNet and MobileNet. It is not clear to me how they are used in the patch gradient context. Are they used as the models parametrized by $\theta_1$? Do they take as input images of size $m \times n$ and outputs a vector of size $s$ that becomes the "pixel" of the latent representation $\mathbf Z$?
- How are the baselines chosen for the experiments? Do they have the same number of parameters / same architecture than the PD models?
- How much memory is gained w.r.t. the number of patches? Could you have a mathematical formula?
- The authors claim that the proposed method is compatible with any existing CNN architecture. However, it seems like the architectures are used as backbones in a more global pipeline? To adapt to segmentation, we would need a new pipeline, we cannot trivially adapt this method to e.g. a U-Net architecture.

# Typos

- Line 121, We => we

---

### Official Review · Reviewer_QrDg · 2023-10-24
**This paper suggests a novel method for training CNN models on large images. Achieved results outperform existing baseline in terms of accuracy. Work looks like lightning talk, not like a finished research. My recommendation: REJECT.**

**Confidence:** 5

**Review:**

## Summary

This paper suggests a novel method for training CNN models on large images. Achieved results outperform the existing baseline in terms of accuracy. The overall idea is to split images into regions, batching them together and processing them separately. Based on reported experiments authors claim improving results compared to gradient-descent method.

## Strengths

1. Writing is easy to follow, language is good
2. High-quality self-explanatory visual elements (like Figure 2)
3. Comparison with alternative methods


## Weaknesses

1. **Important information is missed**
   - no limitations section
   - no risks section
   - no background study
   - no ablation study

   Without this information, it is hard to evaluate the impact of the work on the current state-of-the-art. For example, convolutional networks are known for their locality properties, close pixels construct meaningful objects that being convolved become high-order features used for final decision-making. Splitting image into several patches would break the locality property. Therefore, it would be nice to explain from the data point why is classification result higher - is it due to special features of input images or due to the special attributes of the proposed method, such as position vectors.

2. **No way to reproduce results**
   - no link to code to reproduce reported results
   - no information on hardware setup, base training pipelines
   - no details on the patch_extractor function from Algorithm 2
   - what model is used as theta_1 in the described experiments

   Thus, it allows to question reported results that might bring a wrong impression to future readers. I would highly recommend addressing this item in the future.

3. **No evaluation on training and inference time**
   The proposed method assumes that the input image is patched into M parts and patches are batched. This means that instead of N image, we get N*M images. This should influence training time. Paper misses this evaluation, while training time is not less crucial for efficient training than required memory. The same question goes to the inference stage. How does patching influence inference latency and throughput? Can it be parallelized? Can it be built into the inference model before deployment?

---

### Meta-Review · Area_Chair_5Z6D · 2023-10-27

**Recommendation:** Reject
**Confidence:** 3

**Metareview:**

This paper proposes the idea of PatchGD to seamlessly train deep-learning models on expansive images, by smartly segmenting and updating a core information-gathering element using portions of the image before the final evaluation.

The current form of this paper has unreliable evaluation results, insufficient method details, and missing related work discussion. Though all three reviewers acknowledge/appreciate the methodology innovation, the AC tends to reject this manuscript. Authors should follow reviewers' suggestions to further polish the manuscript.

---

### Decision · Program_Chairs · 2023-10-28

**Decision:**

Accept (Poster)

**Comment:**

All the reviewers confirm the novelty of the idea. One reviewer has raised the concern that not all comparisons are fair enough as they are potentially reported in the settings that are more advantageous for the presented method, which is a valid concern that should not be overlooked. Despite that, the preliminary results reported in the article suggest that PatchGD manages to converge on the set of considered problems, while enabling the training of high-resolution images on the devices with low memory capacity. There are some valid remarks on insufficient method details, and missing related work discussion, that can be addressed in the camera-ready version. We agree with the reviewers that the work has a lot of room for improvement and the method requires a more thorough evaluation. Provided that the authors address the concerns raised by reviewers, we decide to accept the paper based on the actuality of the topic, as memory efficient training is one of the core interests of the workshop. We believe that this work can introduce an interesting discussion at the workshop on the ways to perform the training under memory constraints. Therefore, we also encourage the authors to discuss how their method compares to other memory saving techniques for training neural networks, such as activation checkpointing, offloading and distributed training. Congratulations and hope to see you in person at the workshop and brainstorm on efficient training research together!